# The Role of MECP2 and CCR5 Polymorphisms on the Development and Course of Systemic Lupus Erythematosus

**DOI:** 10.3390/biom10030494

**Published:** 2020-03-24

**Authors:** Ewa Rzeszotarska, Anna Sowinska, Barbara Stypinska, Ewa Walczuk, Anna Wajda, Anna Lutkowska, Anna Felis-Giemza, Marzena Olesinska, Mariusz Puszczewicz, Dominik Majewski, Pawel Piotr Jagodzinski, Michal Czerewaty, Damian Malinowski, Andrzej Pawlik, Malgorzata Jaronczyk, Agnieszka Paradowska-Gorycka

**Affiliations:** 1Department of Molecular Biology, National Institute of Geriatrics, Rheumatology and Rehabilitation, 02-637 Warsaw, Poland; ewa-rzeszotarska1@wp.pl (E.R.); barbara.stypinska@wp.pl (B.S.); ewa.walczuk@spartanska.pl (E.W.); annawajda2046@gmail.com (A.W.); 2Department of Computer Science and Statistics, Poznan University of Medical Sciences, 60-806 Poznan, Poland; ania@ump.edu.pl; 3Department of Biochemistry and Molecular Biology, Poznan University of Medical Sciences, 60-781 Poznan, Poland; alutkowska@op.pl (A.L.); pjagodzi@ump.edu.pl (P.P.J.); 4Department of Connective Tissue Diseases, National Institute of Geriatrics, Rheumatology and Rehabilitation, 02-637 Warsaw, Poland; annafelis@wp.pl (A.F.-G.); marzena.olesinska@vp.pl (M.O.); 5Department of Rheumatology and Internal Diseases, Poznan University of Medical Science, 61-545 Poznan, Poland; mariuszpuszczewicz@gmail.com (M.P.); dmajes@poczta.onet.pl (D.M.); 6Department of Physiology, Pomeranian Medical University, 70-111 Szczecin, Poland; michal.czerewaty@wp.pl (M.C.); pawand@poczta.onet.pl (A.P.); 7Department of Pharmacokinetics and Therapeutic Drug Monitoring, Pomeranian Medical University, 70-111 Szczecin, Poland; malinowski.damian@yahoo.pl; 8Department of Drug Biotechnology and Bioinformatics, National Medicines Institute, 30/34 Chelmska Str., 00-725 Warsaw, Poland; m.jaronczyk@nil.gov.pl

**Keywords:** systemic lupus erythematosus (SLE), polymorphism, methyl-CpG binding protein 2 (MECP2), C-C chemokine receptor type 5 (CCR5)

## Abstract

Systemic lupus erythematosus (SLE) is a chronic and systemic autoimmune disease. SLE is described by production of autoantibodies and causes damage of many organs. T-cells play a crucial role in SLE pathogenesis. T-cells intensify inflammation through a number of processes, which leads to autoimmunization. CCR5 and MECP2 genes are linked with T-cells and pathogenesis of SLE. Polymorphisms in these genes are related with the prognostic factors of risk of disease onset and disease severity. The aim of this study was to estimate the influence of polymorphisms in MECP2 and CCR5 genes on the development and course of systemic lupus erythematosus. We examined 137 SLE patients and 604 healthy controls. We studied polymorphisms for CCR5 gene: rs333 and for MECP2: rs2075596, rs1734787, rs17435, and rs2239464. We genotyped our MECP2 samples and we performed a restriction fragment length polymorphism (RFLP) analysis for CCR5 samples. We showed a risk factor for allele T in rs17435 and for allele A in rs2075596 in MECP2. We noticed that MECP2 rs2075596 G/A, rs1734787 C/A, rs17435 A/T, and rs2239464 G/A polymorphisms are more prevalent in SLE patients than in healthy controls. We believe that above-mentioned MECP2 polymorphisms can be considered as SLE susceptibility factor.

## 1. Introduction

Systemic lupus erythematosus (SLE) is one of the chronic autoimmune, multi-organ diseases with production of autoantibodies and tissues injury [1]. SLE is a disease of poorly understood pathogenesis, clinically heterogeneous and genetically complex, where immune homeostasis and self-tolerance is actively regulated by several types of cells as well as cytokines. An important role in the maintaining and driving of SLE pathogenesis belongs to the T cells. T-cells by secretion of pro-inflammatory cytokines, accumulation of autoreactive T cells as well as helping B-cells in autoantibodies production amplifying inflammation leading to the autoimmunization [2]. T cells from SLE patients show functional and phenotypic anomalies, which may be related with inadequate chemokine receptors expression and/or abnormalities in DNA methylation [2,3,4,5,6,7,8].

Chemokines are involved in the migration and also activation of leukocytes in places of inflammation [9]. C-C chemokine receptor type 5 (CCR5) is chemokine receptor and it’s expressed by cells, such as: macrophages, lymphocytes T, endothelial cells, peripheral blood-derived dendritic cells and others [10,11]. CCR5 is involved in the recruitment of inflammatory cells into tissues. Mechanisms altering expression and function of CCR5 may interfere in development of SLE. This can influence the clinical course of the illness [11].

Non-functional receptor is produced in cause of deletion of 32 bp in CCR5 gene [12]. Some studies show that Δ32 bp deletion in CCR5 gene can be associated with the increased risk of SLE development [3], however, for example Carvalho et al. [11] suggests that association between SLE and CCR5Δ32 is protective. Also Schauren et al. [4] found that in European-derived patients with this disease deletion of 32 bp may be protective factor against SLE development.

Methyl-CpG binding protein 2, X- chromosome protein coding MECP2 gene is considered as a systemic lupus erythematosus genetic factor, because of altered regulation of T-cells genes, which are sensitive for methylation and also the fact, that SLE is more common in women. MECP2 participate in epigenetic transcriptional regulation of genes, which are sensitive for methylation. MECP2 is considered as a gene for SLE, because of his pivotal role in transcriptional suppression of genes sensitive for methylation and because the fact, that genes sensitive to methylation DNA are overexpressed in SLE [5,6,7,13]. Some polymorphisms within MECP2 gene, such as: rs17435, rs2075596, rs3027933, rs1624766, rs1734787, rs1734791, rs1734792, and rs2239464, were found as associated with SLE [5,6,7,8].

Polymorphisms within the CCR5 and MECP2 genes are associated with the risk of SLE disease onset and they may be also a prognostic factors of SLE severity [7,8,14]. Research in these subjects can improve the diagnosis and prognosis of SLE, which could result in faster and more effective treatment of SLE patients. That is why we decide to assess the impact of polymorphisms in the CCR5 and MECP2 genes on the development and course of systemic lupus erythematosus in the Polish population.

## 2. Materials and Methods

### 2.1. Study Group

One hundred thirty seven patients with SLE and 604 healthy controls were included in this study. SLE patients were recruited from the National Institute of Geriatrics, Rheumatology and Rehabilitation in Warsaw, Poland and from the Poznan University of Medical Sciences, Poland. All patients with SLE included in study met the classification criteria of American College of Rheumatology (ACR) for this disease (SLE patients included in the study have the presence of at least four criteria). The control group did not reveal any (laboratory or clinical) signs of autoimmunology diseases. All patients and control subjects had the same origin and socioeconomic status. All subjects participating in the present study provided written informed consent for genetic studies and consented to sample collection that were approved by the Research Ethics Committee of the National Institute of Geriatrics, Rheumatology and Rehabilitation in Warsaw and by the Research Ethics Committee of the Poznan University of Medical Sciences. This study conducted in accordance with the ethical standards of our Institute and with the 1964 Helsinki declaration and its later amendments or comparable ethical standards.

### 2.2. Methods

#### 2.2.1. SNP Selection

Single nucleotide polymorphisms (SNPs) selection was preceded by research in PubMed. SNPs selected for this study are recorded in the public dbSNP database. SNPs have been chosen due to their significant association with autoimmune diseases, including SLE and potential clinical significance. MECP2 SNPs with MAF (Minor Allele Frequency) below 5% (< 0.05) were excluded from study. We selected four MECP2 genetic variants (rs2075596, rs1734787, rs17435, and rs2239464) and one for CCR5—rs333.

#### 2.2.2. DNA Extraction

Whole blood samples obtained from the SLE patients and controls were collected in EDTA tubes. DNA was extracted from peripheral blood by using QIAamp DNA Blood Mini Kit (Qiagen, Hilden, Germany).

#### 2.2.3. Genotyping

SNP genotyping assay was performed on QuantStudio 5 (Applied Biosystems, Forester City, CA, USA) in accordance with the conditions recommended by the manufacturer. TaqMan SNP genotyping assays were used to genotype SNPs of MECP2: C__15765472_10 (rs2075596), C___8966344_1_ (rs1734787), C___2597094_20 (rs17435), C___2277191_1_ (rs2239464) in samples. For CCR5 we carried out a Restriction Fragment Length Polymorphism (RFLP) analysis. We selected restriction enzymes basing on article [15]. Length of the restriction fragments was 189 bp for HOM allele and 157 bp for HOMΔ32 allele. The accuracy of genotyping of MECP2 was confirmed by Sanger sequencing using an ABI PRISM 3130 Genetic Analyzer (Applied Biosystems, Forester City, CA, USA).

### 2.3. Statistical Analysis

The clinical data, disease activity and laboratory variables in relation to MECP2 and disease activity and laboratory variables in relation to CCR5 were described as mean ± standard deviation or median with range. The odds ratios (OR) were calculated. The analysis was carried out under 4 genetic models (codominant, dominant, recessive, and overdominant). A *p*-value lower than or around 0.05 (*p* < 0.05) was considered statistically significant. The links between experienced SNPs and disease activity parameters were examined using the Mann–Whitney, Cochran Cox and T-student tests. Bonferroni correction was used to adjusting *p*-values for multiple testing; this correction was used to determine the association between examined gene polymorphisms and clinical phenotype of SLE. Bonferroni-corrected α-level of *p* < 0.003 was considered statistically significant. Computational functional prediction analysis was performed using MutationTaster 2.0 [16] and PROVEAN v.1.1.3 [17] programs to study the influence of the examined SNPs on the function of associated protein.

## 3. Results

### 3.1. Patients Characteristics

The patients clinical and biochemical data were collected at the time blood sampling and these information are summarized in Table 1. The SLE patients were in the active stage of disease with mean SLE disease activity index (SLEDAI) > 4 and with the mean disease duration 7 years. A several of different blood autoantibody is present in our SLE patients. We observed that the most frequent autoantibody, present in 74% of our SLE patients, was anti-dsDNA, while the less frequent autoantibody, present in 2% of our patients with SLE was anti-Jo.

### 3.2. MECP2 and CCR5 Polymorphisms and SLE Susceptibility

First we investigated the prevalence of genotypes in MECP2: rs2075596 G/A, rs1734787 C/A, rs17435 A/T, rs2239464 G/A, and CCR5 rs333 genes in SLE patients and healthy controls to find out if there is an association with specific genotype and/or allele and the occurrence of the SLE disease. The distributions of genotypes and allele frequencies of MECP2 and CCR5 polymorphisms among SLE patients and healthy controls with their relations with the risk of SLE diseases are presented in Table 2.

The minor allele frequency (MAF) of the four chosen MECP2 SNPs: rs2075596, rs1734787, rs17435 and rs2239464 in our study were comparable to the European ancestry (1000Genomes database; Appendix A). MECP2 and CCR5 minor allele frequency values in Polish SLE patients generally was higher than in controls and in European population (1000Genomes). Only in CCR5 (rs333 32 bp deletion allele) MAF of controls is higher than in SLE patients. MAF of MECP2 is equal as control just in one case (MECP2 rs2075596 A allele).

In the case of MECP2 rs2075596 G/A, only dominant model (GA + AA) in comparison SLE patients with healthy controls is statistically significant (*p* = 0.047). This genotype model occurs more often in SLE patients (34.81%) than in control group (25.83%), odds ratio (OR) is 1.534.

In the case of MECP2 rs1734787 C/A, two models in comparison SLE patients with healthy controls are statistically significant. First is codominant model with CA genotype (*p* = 0.030) and second is overdominant model with CA genotype (*p* = 0.018). Both genotype models occur more often in SLE patients (27.8%) than in control group (17.2%). Odds ratio (OR) in codominant model are 1.796 and 1.847; respectively.

In the case of MECP2 rs17435 A/T, three models in comparison SLE patients with healthy controls are statistically significant. First is codominant model with AT genotype (*p* < 0.00001), second is dominant model with AT+TT genotype (*p* = 0.0001) and third is overdominant model with AT genotype (*p* < 0.00001). All three genotypes mentioned above occur more frequent in SLE patients than in control group. The occurrence of the genotype AT in first, codominant model in SLE patients is 39.4% and in control group is 15.6%, odds ratio (OR) is 3.494. The occurrence of the genotype AT + TT in second, dominant model in SLE patients is 48.9% and in control group is 29.6%, odds ratio (OR) is 2.279. The occurrence of the genotype AT in third, overdominant model in SLE patients is 39.4% and in control group is 15.6%, odds ratio (OR) is 3.534.

In the case of MECP2 rs2239464 G/A, three models in comparison SLE patients with healthy controls are statistically significant. First is codominant model with GA genotype (*p* = 0.00004), second is dominant model with GA+AA genotype (*p* = 0.005) and third is overdominant model with GA genotype (*p* < 0.00001). All three genotypes mentioned above occur more frequent in SLE patients than in control group. The occurrence of the genotype GA in first, codominant model in SLE patients is 39.8% and in control group is 19%, odds ratio (OR) is 2.665. The occurrence of the genotype GA + AA in second, dominant model in SLE patients is 46.1% and in control group is 31.6%, odds ratio (OR) is 1.850. The occurrence of the genotype GA in third, overdominant model in SLE patients is 39.8% and in control group is 19%, odds ratio (OR) is 2.830.

In CCR5 case odds ratio (OR) and genotype models in comparison patients with SLE with healthy controls are not statistically significant, what is shown in Table 2. We also did the bioinformatic analysis to check if examined SNPs affect the function of the associated protein. Computational functional prediction analysis was performed using MutationTaster and PROVEAN programs. The analysis showed that not MECP2 SNPs, but deletion of 32 bp in rs333 CCR5 might result in altered function of the associated protein. The amino acid sequence is changed, what consequences in frameshift. It leads to changes in splice site and also truncate the protein (might cause nonsense-mediated mRNA decay (NMD)).

Our analysis have shown that MECP2 gene rs2075596 A allele and rs17435 T allele were associated with significantly increased risk of SLE (*p* = 0.018 and *p* = 0.021, respectively) in our Polish population.

### 3.3. MECP2 SNPs and SLE Phenotype

Because MECP2 polymorphisms revealed an association in the pooled analysis of Polish subjects, and because of we hypothesize that MECP2 gene and/or CCR5 gene may be good candidate genes to play a part in SLE pathogenesis, we decided to carry on analysis whether genetic variants located in these genes may have an impact on SLE phenotype. A detailed genotype-phenotype was conducted among SLE patients with combined genotypes under the dominant and the recessive models for each examined polymorphisms in relation to clinical and laboratory parameters.

#### 3.3.1. MECP2 rs2075596 G/A Polymorphism

Next, we compared disease activity and laboratory parameters in dominant model GA+AA versus GG genotype in MECP2 rs2075596 G/A, what is shown in Table 3. Without Bonferroni correction our genotype-phenotype have shown that in dominant model GA+AA versus GG in MECP2 rs2075596 G/A only median age of SLE patients, can be considered as statistically significant (*p* = 0.053). Patients with rs2075596 G allele have earlier age of disease onset (37.5 years) than SLE patients with rs2075596 A allele. However, after Bonferroni correction there was no significant association between MECP2 rs2075596 G/A polymorphism and SLE phenotype.

#### 3.3.2. MECP2 rs1734787 C/A Polymorphism

Next, we compared disease activity and laboratory parameters in dominant model CC+CA versus AA genotype in MECP2 rs1734787 C/A, what is shown in Table 4. Without Bonferroni correction we observed that: a) in dominant model in MECP2 rs1734787 C/A median age of SLE patients, which is 30.5 years in CC+CA genotype and 39 years in AA genotype is statistically significant (*p* = 0.02), b) patients with allele A have statistically significant (*p* = 0.04) much more value of ESR, which is 13.5 mm/h than those with allele C with ESR of 0.7 mm/h, c) however, patients with allele C in comparison with allele A have much more level of creatinine (28 vs. 0.98 mg/dL, *p* = 0.01), ALT (38 vs. 20 U/L, *p* = 0.04) and AST (38.62 vs. 23.17 U/L; *p* = 0.01). INR values are elevated in patients with allele C. After Bonferroni correction no association could be detected between the MECP2 rs1734787 C/A variant and SLE phenotype.

#### 3.3.3. MECP2 rs17435 A/T Polymorphism

Afterwards, we compared disease activity and laboratory parameters in recessive model AA+AT versus TT genotype in MECP2 rs17435 A/T, what is shown in Table 5. Patients with allele T have statistically significant (*p* = 0.04) much more value of ESR, which is 13.5 mm/h than those with allele A with ESR of 0.7 mm/h. However, patients with allele A in comparison with allele T have much more level of creatinine (28 vs. 0.98 mg/dL, *p* = 0.01), ALT (20 vs. 38 U/L, *p* = 0.04) and AST (38.62 vs. 23.17 U/L; *p* = 0.01). INR values are elevated in patients with allele C. However, after Bonferroni correction no association could be detected between the MECP2 rs17435 A/T genetic variant and SLE phenotype.

#### 3.3.4. MECP2 rs2239464 G/A Polymorphism

Subsequently, we compared disease activity and laboratory parameters in recessive model GG+GA versus AA genotype in MECP2 rs2239464 G/A, what is shown in Table 6. After Bonferroni correction our analysis indicate significant association of the MECP2 rs2239464 G/A SNP with mean value of ESR level and mean value of ALT level as well as we also association with tendency between MECP2 rs2239464 G/A and mean value of creatine level and mean value of AST level. Patients with AA genotype have statistically (*p* = 0.034) higher age, which is 45.5 years than patients with GG+GA genotype, which is 31 years. Patients with allele A have also statistically significant (*p* = 0.003) much more value of ESR, which is 21 mm/h than those with allele G with ESR of 0.7 mm/h. However, patients with allele G in comparison with allele A have statistically significant higher level of creatinine (28 vs. 0.955 mg/dL; *p* = 0.007), ALT (42.54 vs. 18.25 U/L; *p* = 0.002) and AST (38.62 vs. 20.75 U/L; *p* = 0.004).

### 3.4. CCR5 Polymorphism

Furthermore, we compared disease activity and laboratory parameters in recessive model HET + HOM∆32 versus HOM genotype in rs333, what is shown in Table 7. Our data have shown that examined CCR5 gene polymorphism do not have a high impact on SLE risk as well as SLE phenotype. In patients with HET+HOM∆32 versus HOM genotype in CCR5 gene concentration of TG is 45.67 mg/dL, where in HOM this value is 58.83 mg/dL, what can be considered as significant (*p* = 0.06). In patients with HET+HOM∆32 genotype concentration of AST is 25 (U/L), where in HOM this value is 31.5 (U/L), what is considered as significant (*p* = 0.02). However, after Bonferroni correction above association are not significant.

## 4. Discussion

Systemic lupus erythematosus is a multifactorial, chronic autoimmune disease, characterized by the production of autoantibodies, mostly those directed against nuclear antigens. This leads to tissue inflammation and eventually impairment of organs [18,19].

In this study we examined the impact of polymorphisms within MECP2 and CCR5 gene on the development and course of SLE. The knowledge about factors, which impact on systemic lupus erythematosus have a crucial role on better understanding of this disease. Searching for new polymorphisms associated with SLE can improve diagnosis and prognosis of SLE. Polymorphisms within examined genes are associated with prognostic factors of disease severity and the risk of disease onset.

In the present study, we observed that the MECP2 rs2075596 A allele, rs17435 T allele, rs1734787 CA genotype, and rs2239464 GA genotype have shown a significant association with risk of SLE. Additionally, in our study, we have also observed that the MECP2 rs1734787 CC genotype, rs17435 A allele and rs2239464 SNP may showed a positive association with disease severity. Our research exposed that polymorphisms within MECP2 and CCR5 genes may have an impact on elevated ALT and AST values. Alanine aspartate aminotransferase (AST) and alanine aminotransferase (ALT) are enzymes, that are biomarkers and they are measured to assess liver health [20].

Liver involvement in SLE disease is observed in the literature [21,22,23,24]. While some researchers claim that clinical liver dysfunction is not common in SLE [21,25], others say that it is frequent [22,26]. Regardless of this, the fact is that from 9.8 (in juvenile patients [27]) to 59.7% of patients with this disease may have abnormalities in liver functioning in their lifes [21,22,23,25]. Takashaki et al. [22] showed that liver dysfunction is found in 59.7% of SLE patients. Runyon et al. [23] found that 55% of SLE patients had SLE and liver disease in the same year. The liver dysfunction is multifactorial and can be caused by drug intake, SLE disease, autoimmune hepatis (AIH), fatty liver, cholangitis, primary biliary cholangitis (PBC), and viral hepatis [21,22,24,26].

MECP2 and CCR5 genes are involved in pathogenesis of liver diseases. As it was tested [28], in the patients’ livers with hepatic cirrhosis levels of CCR5 mRNA were significantly raised up. What is more, CCR5 ligand RANTES was much higher in patients with this disease [28]. Stimulation with RANTES causes increased proliferation and migration of HSCs (hepatic stellate cells), two pivotal elements of the fibrogenic process [29]. Likewise, MECP2 gene is an element from epigenetic relay pathways, which regulate hepatic fibrogenesis. MECP2 participates in Hedgehog and Wnt signaling in liver fibrosis [30,31]. MECP2 suppresses PPARγ (Peroxisome Proliferator-Activated Receptor-gamma), what is needed to converse HCSs into myofibroblasts [32]. PPARγ negatively controls expression of Col1a1 (type I collagen) [31,32]. Increased level of Col1a1 in Hepatic Stellate Cells causes the presence of hepatic fibrosis [31].

Accordingly, polymorphisms within the MECP2 gene (rs1734787 and rs2239464) may be associated with earlier disease onset and more severe course of the disease. However, rs2075596 polymorphism might be connected only with earlier disease onset. Also, polymorphism within MECP2 gene (rs17435) and homozygote genotype for CCR5 can be linked only with more severity of the SLE. There is little research in the literature specific MECP2 polymorphisms with SLE activity or severity. Alesaidi et al. [33] demonstrated no significant association between rs1734791, rs1734787 in MECP2 and clinical features. Doudar et al. [34] also did not detect significant relationship between rs1734791 in MECP2 and disease severity or activity. In Sanchez et al. [35] study rs17435 in MECP2 was not associated with clinical manifestation of SLE. However, in Carmona et al. [36] research rs17435 is linked with more severe course of other autoimmune disease, diffuse cutaneous systemic sclerosis (dcSSc), the most severe form of systemic sclerosis (SSc).

Joyner et al. [37] found an association between rs2239464 polymorphism in MECP2 and reduced cortical surface area in brain in two different human populations.

Rossol et al. [38] showed influence of CCR5Δ32 polymorphism on the clinical course of rheumatoid arthritis. Deletion of 32 bp of CCR5 is protective for joint inflammation and other manifestations of the disease. Protective effect of this deletion also is revealed by Pokorny et al. [39]. Nonetheless, John et al. [40] did not show influence of CCR5 polymorphism on clinical course of RA. This is supported by Lindner et al. [41] in RA and juvenile idiopathic arthritis (JIA) patients.

Meta-analysis done by Liu at al. [42] showed that may be a SLE risk factor for allele A in rs2075596 and rs2239464 in the MECP2 gene. Our study confirmed a risk factor for allele A in rs2075596 and revealed this for allele T in rs17435 of the MeCP2 gene.

We didn’t observe more prevalence of SLE patients with polymorphisms within CCR5 gene. The same observation was made by Aguilar et al. [43] and Martens et al. [44]. Aguilar et al. [43] found no association between deletion of 32 bp in CCR5 gene and SLE susceptibility in Spanish patients. Martens et al. [44] also confirmed a lack of association between CCR5Δ32 polymorphism and SLE susceptibility. Schauren et al. [4] found in European-derived patients in Brazil that heterozygotes with CCR5Δ32 allele are less common in SLE patients than in controls. It follows that CCR5Δ32 allele might be a protective allele for SLE in patients. The same conclusion was made by Carvalho et al. [11] in Portuguese patients. In contrast, Baltus et al. [14] claim that CCR5Δ32 polymorphism can be considered as a risk factor in female Brazilian patients, and it is linked to the age of SLE onset. Furthermore, Mamtani et al. [3] showed that CCR5 haplotype HHG*2, which contains CCR5Δ32 polymorphism is connected with an elevated risk of developing systemic lupus erythematosus. However, research on the prevalence of CCR5 polymorphisms in SLE patients are ambiguous. The noticeable divergences in role of CCR5 polymorphisms in SLE patients can be explained by different methodology and methods used for testing [14].

We noticed that the below mentioned polymorphisms like rs2075596 G/A, rs1734787 C/A, rs17435 A/T and rs2239464 G/A within MECP2 gene are more widespread in SLE patients than in healthy controls. These observations are consistent with those made by Alesaeidi et al. [33] in Iranian population, Webb et al. [7] in European, Sawalha et al. [5] in Korean and also by Kaufman et al. [8] in four different populations. Bentham et al. revealed in his GWAS analysis that MECP2 rs1734787 is an appropriate SLE risk factor for people of European ancestry [45]. However, Dong et al. [46] in Chinese Northern Han revealed that GG genotype in rs2075596 and rs2239464 in MECP2 can protect against SLE.

We have to admit that our study has limitations. First is limited number of SNPs, which were examined in this study. Second, our sample size may be too small to observe some connections. Nevertheless, our samples are characterized carefully due to serology and clinical phenotype. In addition, they originate from mono-ethnic population.

## 5. Conclusions

In conclusion, we confirmed the role of genetic polymorphism within MECP2 gene on SLE susceptibility and severity. The present study showed a risk factor for allele A in rs2075596 and allele T in rs17435 of the MECP2 gene. We also noticed a connection between some polymorphisms, development and course of systemic lupus erythematosus. We confirmed that some polymorphisms, such as: rs2075596, rs1734787, rs17435 and rs2239464 within MECP2 gene are more prevalent in SLE patients than in healthy controls.

Our research established that these polymorphisms within MECP2 can be considered as genetic susceptibility factor for systemic lupus erythematosus. Moreover, we believe that the abovementioned polymorphisms can be predictive factors for the development and course of the SLE. There is a need for further research with larger sample size, especially more different ethnic populations worldwide. Additionally, we believe that our outcomes can be helpful in diagnosis and prognosis the course of SLE.

## Figures and Tables

**Table 1 biomolecules-10-00494-t001:** Clinical characteristics of the study cohorts.

Characteristics	SLE Patients	
	N*	Mean Values (Range)
Age [years]	100	41 (21–87)
Disease duration [years]	63	7 (0–43)
SELENA_SLEDAI	63	4 (0–26)
BMI	58	24.2 (18.3–36.3)
Hemoglobin [g/dL]	58	12.3 (4.9–16)
PLT [×10^3^/mm^3^]	58	214.5 (38–598)
APTT	48	31.85 (21.3–127)
PT	47	11 (8.6–100)
INR	48	1 (0.7–3.3)
ESR [mm/h]	55	0.8 (0.5–86)
Creatinine	50	32 (0.6–296)
Urea	57	13 (1–296)
CRP [mg/L]	63	0 (0–30)
C3	62	0 (0–31)
C4	62	1 (0–31)
ALT [U/L]	58	37 (9–317)
AST [U/L]	58	28 (14–850)
Cholesterol	46	114.22 ± 38.41
TG [mg/dL]	46	57.83 ± 21.62
	**N***	**n** (%)**
anti-SSA	59	25 (42.37)
anti-SSB	58	5 (8.62)
anti-Sm	68	20 (29.41)
anti-Rib	57	4 (7.02)
anti-Jo	57	1 (1.75)
anti-Scl70	78	21 (26.92)
anti-CENP-B	58	2 (3.45)
anti-U1RNP	73	30 (41.09)
anti-IgM	59	6 (10.17)
anti-IgG	60	17 (28.33)
anti-dsDNA	93	69 (74.19)

N*—number of patients with clinical information’s; n**—number of patients with positive clinical manifestation; TG—triglyceride; AST—glutamic oxoloacetic transaminase; ALT—glutamic pyruvic transferase; C3 and C4—complement; CRP—C-reactive protein; ESR—erythrocyte sedimentation ratio; INR—international normalized ratio; PT—prothrombin time; APTT—activated partial thromboplastin time; PLT—platelets; BMI—body mass index; SELENA_SLEDAI—SLE disease activity index.

**Table 2 biomolecules-10-00494-t002:** Distribution of genotypes and allele frequencies of MECP2 and C-C chemokine receptor type 5 (CCR5) polymorphisms among patients with systemic lupus erythematosus (SLE) and healthy subjects (*p* = SLE vs. controls).

MECP2 rs2075596 G/A	Genotype	SLE N (%)	Controls N (%)*	OR (95% CI)	*p* Value
**Codominant**	GG	88 (65.19)	448 (74.17)	reference	
	GA	39 (28.89)	140 (23.18)	1.418 (0.903–2.198)	NS
	AA	8 (5.93)	16 (2.65)	2.545 (0.912–6.524)	0.07
**Dominant**	GG	88 (65.19)	448 (74.17)	reference	
	GA + AA	47 (34.81)	156 (25.83)	1.534 (1.005–2.32)	**0.047**
**Recessive**	GG + GA	127 (94.07)	588 (97.35)	reference	
	AA	8 (5.93)	16 (2.65)	2.315 (0.838–5.874)	NS
**Overdominant**	GG + AA	96 (71.11)	464 (76.82)	reference	
	GA	39 (28.89)	140 (23.18)	1.346 (0.861–2.076)	NS
**Alleles**	G	215 (79.63)	1036 (85.76)	reference	
	A	55 (20.37)	172 (14.24)	1.541 (1.078–2.179)	**0.018**
**MECP2 rs1734787 C/A**	**Genotype**	**SLE** **N (%)**	**Controls** **N (%)***	**OR (95% CI)**	***p* value**
**Codominant**	CC	84 (66.7)	259 (73.2)	references	
	CA	35 (27.8)	61 (17.2)	1.796 (1.054–2.937)	**0.030**
	AA	7 (5.6)	34 (9.6)	0.635 (0.229–1.528)	NS
**Dominant**	CC	84 (66.7)	259 (73.2)	references	
	CA + AA	42 (33.3)	95 (26.8)	1.363 (0.854–2.156)	NS
**Recessive**	CC + CA	119 (94.4)	320 (90.4)	references	
	AA	7 (5.6)	34 (9.6)	0.554 (0.202–1.315)	NS
**Overdominant**	CC + AA	91 (72.2)	293 (82.8)	references	
	CA	35 (27.8)	61 (17.2)	1.847 (1.107–3.048)	**0.018**
**Alleles**	C	203 (80.6)	579 (81.8)	references	
	A	49 (19.4)	129 (18.2)	1.083 (0.735–1.580)	NS
**MECP2 rs17435 A/T**	**Genotype**	**SLE** **N (%)**	**Controls** **N (%)****	**OR (95% CI)**	***p* value**
**Codominant**	AA	70 (51.1)	231 (70.4)	references	
	AT	54 (39.4)	51 (15.6)	3.494 (2.129–5.719)	**< 0.00001**
	TT	13 (9.5)	46 (14.0)	0.933 (0.437–1.880)	NS
**Dominant**	AA	70 (51.1)	231 (70.4)	references	
	AT + TT	67 (48.9)	97 (29.6)	2.279 (1.479–3.506)	**0.0001**
**Recessive**	AA + AT	124 (90.5)	282 (86.0)	references	
	TT	13 (9.5)	46 (14.0)	0.643 (0.307–1.264)	NS
**Overdominant**	AA + TT	83 (60.6)	277 (84.4)	references	
	AT	54 (39.4)	51 (15.6)	3.534 (2.183–5.706)	**<0.00001**
**Alleles**	A	194 (70.8)	513 (78.2)	references	
	T	80 (29.2)	143 (21.8)	1.479 (1.058–2.059)	**0.021**
**MECP2 rs2239464 G/A**	**Genotype**	**SLE** **N (%)**	**Controls** **N (%)***	**OR (95% CI)**	***p* value**
**Codominant**	GG	69 (53.9)	238 (68.4)	references	
	GA	51 (39.8)	66 (19.0)	2.665 (1.648–4.293)	**0.00004**
	AA	8 (6.3)	44 (12.6)	0.627 (0.244–1.433)	NS
**Dominant**	GG	69 (53.9)	238 (68.4)	references	
	GA + AA	59 (46.1)	110 (31.6)	1.850 (1.194–2.857)	**0.005**
**Recessive**	GG + GA	120 (93.7)	304 (87.4)	references	
	AA	8 (6.3)	44 (12.6)	0.461 (0.182–1.029)	0.06
**Overdominant**	GG + AA	77 (60.2)	282 (81.0)	references	
	GA	51 (39.8)	66 (19.0)	2.830 (1.767–4.510)	**< 0.00001**
**Alleles**	G	189 (73.8)	542 (77.9)	references	
	A	67 (26.2)	154 (22.1)	1.248 (0.881–1.756)	NS
**CCR5**	**Genotype**	**SLE** **N (%)**	**Controls** **N (%)****	**OR (95% CI)**	***p* value**
**Codominant**	HOM	106 (78.52)	434 (78.48)	reference	-
	HET	26 (19.26)	113 (20.43)	0.942 (0.561–1.542)	NS
	HOM∆32	3 (2.22)	6 (1.08)	2.047 (0.326–9.756)	NS
**Dominant**	HOM	106 (78.52)	434 (78.48)	reference	-
	HET + HOM∆32	29 (21.48)	119 (21.52)	0.998 (0.607–1.603)	NS
**Recessive**	HOM + HET	132 (97.78)	547 (98.92)	reference	-
	HOM∆32	3 (2.22)	6 (1.08)	2.072 (0.331–9.840)	NS
**Overdominant**	HOM + HOM∆32	109 (80.74)	440 (79.57)	reference	-
	HET	26 (19.26)	113 (20.43)	0.929 (0.554–1.518)	NS

The *p*-values marked in bold are significant. *p* < 0.05 was significant, NS—not significant.

**Table 3 biomolecules-10-00494-t003:** Disease activity and laboratory variables in relation to MECP2 rs2075596 G/A, dominant model.

Parameter	GG	GA + AA	*p*
	Mean ± SD (Median)		
Age (years)	37.5 (21–76)	49.5 (25–87)	0.05 ^a^
Disease duration (years)	8 (1–43)	3 (0–34)	0.7 ^a^
SELENA_SLEDAI	4 (0–16)	8 (0–18)	0.2 ^a^
BMI	24.35 (18.3–36.3)	22.8 (21.8–34.9)	1.0 ^a^
Hemoglobin (g/dL)	12.7 (4.9–15.4)	11.8 (8.9–16)	0.3 ^a^
PLT (× 10^3^/mm^3^)	204.5 (38–598)	223 (50–557)	0.7 ^a^
APTT	32 (21.3–127)	30.8 (24.1–42)	0.6 ^a^
Pt	11.15 (8.6–80)	10.3 (8.7–23.3)	0.4 ^a^
INR	1 (0.8–3.3)	0.9 (0.7–2.2)	0.1 ^a^
ESR (mm/h)	0.8 (0.5–49)	0.9 (0.5–86)	0.1^a^
Creatinine (mg/dL)	33.4 (0.7–296)	32 (0.8–80)	0.7 ^a^
Urea	13 (1–395)	8 (3–163)	0.3 ^a^
CRP (mg/L)	0 (0–8)	0 (0–20)	0.4 ^a^
C3	0 (0–127)	1 (0–53)	0.3 ^a^
C4	1 (0–31)	1 (0–4)	0.5 ^a^
ALT (U/L)	42 (9–317)	27 (15–159)	0.1 ^a^
AST (U/L)	28.5 (14–850)	28 (20–136)	0.5 ^a^

a—Mann–Whitney.

**Table 4 biomolecules-10-00494-t004:** Disease activity and laboratory variables in relation to MECP2 rs1734787 C/A, dominant model.

Parameter	CC + CA	AA	*p*
	Mean ± SD (Median)		
Age (years)	30.5 (18–87)	39 (29–62)	**0.02^c^**
Disease duration (years)	11.46 ± 9.32	7.86 ± 7.52	0.4^a^
SELENA_SLEDAI	5 ± 3.74	9.86 ± 9.77	0.2^b^
BMI	25.23 ± 4.61	23.44 ± 1.92	0.4^a^
Hemoglobin (g/dL)	13.2 4.9–15.3	10.75 8.9–13.3	0.2^c^
PLT (× 10^3^/mm^3^)	190 (111–598)	256 (162–455)	0.3^c^
APTT	34 (24–127)	25.85 (24.2–27.5)	0.1.^c^
Pt	11.4 (10–34)	10.3 (10–11.1)	0.2.^c^
INR	1.05 (0.9–3.3)	0.9 (0.89–0.98)	**0.06^c^**
ESR (mm/h)	0.7 (0.5–1.3)	13.5 (0.5–86)	**0.04^c^**
Creatinine (mg/dL)	28 (20–78)	0.98 (0.84–24)	**0.01^c^**
Urea	15 (2–262)	27.2 (3–41)	0.9^c^
CRP (mg/L)	1 (0–2)	1 (0–30)	0.6^c^
C3	1 (0–67.8)	1 (0–53)	0.3^c^
C4	1 (0–7.83)	1 (0–12)	0.8^c^
ALT (U/L)	38 (9–79)	20 (15–42)	**0.04^c^**
AST (U/L)	38.62 ± 17.75	23.17 ± 5.53	**0.01^c^**
ALP	69 (40–316)	67.5 (65–70)	0.9^c^

A—T-student, b—Cochran Cox, c—Mann–Whitney. The *p*-values marked in bold are significant.

**Table 5 biomolecules-10-00494-t005:** Disease activity and laboratory variables in relation to MECP2 rs17435 A/T, recessive model.

Parameter	AA + AT	TT	*p*
	Mean ± SD (Median)		
Age (years)	31 (18–87)	33 (21–62)	0.3^c^
Disease duration (years)	11.46 ± 9.32	7.86 ± 7.51	0.1^a^
SELENA_SLEDAI	5 ± 3.74	9.86 ± 9.77	0.2^b^
BMI	25.23 ± 4.61	23.44 ± 1.92	0.4^a^
Hemoglobin (g/dL)	13.2 (4.9–15.3)	10.75 (8.9–13.3)	0.2^c^
PLT (× 10^3^/mm^3^)	190 (111–598)	256 (162–455)	0.3^c^
APTT	34 (24–127)	25.85 (24.2–27.5)	0.1^c^
Pt	11.4 (10–34)	10.3 (10–11.1)	0.2^c^
INR	1.05 (0.9–3.3)	0.9 (0.89–0.98)	**0.05^c^**
ESR (mm/h)	0.7 (0.5–1.3)	13.5 (0.5–86)	**0.04^c^**
Creatinine (mg/dL)	28 (20–78)	0.98 (0.84–24)	**0.01^c^**
Urea	15 (2–262)	27.2 (3–41)	0.9^c^
CRP (mg/L)	1 (0–2)	1 (0–30)	0.6^c^
C3	(0–67.8)	1 (0–53)	0.3^c^
C4	1 (0–7.83)	1 (0–12)	0.8^c^
ALT (U/L)	38 (9–79)	20 (15–42)	**0.04^c^**
AST (U/L)	38.62 ± 17.75	23.17 ± 5.53	**0.01^b^**
ALP	69 (40–316)	67.5 (65–70)	0.9^c^

a—T-student, b—Cochran Cox, c—Mann–Whitney, The *p*-values marked in bold are significant.

**Table 6 biomolecules-10-00494-t006:** Disease activity and laboratory variables in relation to MECP2 rs2239464 G/A, recessive model.

Parameter	GG + GA	AA	*p*
	Mean ± SD (Median)		
Age (years)	31 (18–72)	45.5 (32–62)	**0.03^c^**
Disease duration (years)	14 (1–30)	12 (2–20)	1.0^c^
SELENA_SLEDAI	5 ± 3.74	13.2 ± 9.65	0.1^b^
BMI	25.23 ± 4.61	24.37 ± 2.03	0.8^a^
Hemoglobin (g/dL)	13.2 (4.9–15.3)	11.1 (8.9–13.3)	0.3^c^
PLT (× 10^3^/mm^3^)	190 (111–598)	324 (162–455)	0.2^c^
APTT	34 (24–127)	24.2 (24.2–24.2)	1.0^c^
Pt	11.4 (10–34)	10.7 (10.3–11.1)	0.5^c^
INR	1.05 (0.9–3.3)	0.935 (0.89–0.98)	0.1^c^
ESR (mm/h)	0.7 (0.5–1.3)	21 (9–86)	**0.003^c^**
Creatinine (mg/dL)	28 (20–78)	0.955 (0.84–1.21)	**0.007^c^**
Urea	15 (2–262)	36.7 (18–41)	0.2^c^
CRP (mg/L)	1 (0–2)	7 (0–30)	0.1^c^
C3	1 (0–67.8)	39 (0–53)	0.1^c^
C4	1 (0–7.83)	1 (0–12)	0.9^c^
ALT (U/L)	42.54 ± 22.25	18.25 ± 4.27	**0.002^b^**
AST (U/L)	38.62 ± 17.75	20.75 ± 3.30	**0.004^b^**
ALP	69 (40–316)	70 (70–70)	1.0^c^

a—T-student, b—Cochran Cox, c—Mann–Whitney, The *p*-values marked in bold are significant.

**Table 7 biomolecules-10-00494-t007:** Disease activity and laboratory variables in relation to CCR5 rs333, recessive model.

Parameter	HOM	HET + HOM∆32	*p*
	Mean ± SD (Median)		
Age (years)	43 (21–87)	37 (21–66)	0.3^a^
Disease duration (years)	8 (0–43)	4 (1–23)	0.8^a^
SELENA_SLEDAI	4 (0–18)	6 (0–16)	0.4^a^
BMI	24.3 (18.3–36.3)	24.1 (19–31.1)	0.6^a^
Hemoglobin (g/dL)	12.65 (4.9–16)	12 (9.9–15.4)	1.0^a^
PLT (× 10^3^/mm^3^)	203 (38–596)	252 (95–598)	0.2^a^
APTT	32 (21.3–78.4)	26 (24.1–127)	0.3^a^
Pt	10.95 (9.6–80)	10.4 (8.6–34)	0.2^a^
INR	1 (0.8–2.8)	0.98 (0.7–3.3)	0.1^a^
ESR (mm/h)	0.8 (0.5–86)	0.75 (0.5–9)	0.2^a^
Creatinine (mg/dL)	33.2 (0.8–296)	33 (0.7–80)	0.9^a^
Urea	12.5 (1–395)	12.5 (3–63)	0.8^a^
CRP (mg/L)	0 (0–20)	0 (0–2)	0.9^a^
C3	0 (0–127)	1 (0–113)	0.3^a^
C4	1 (0–31)	1 (0–17.5)	0.3^a^
ALT (U/L)	41 (9–317)	30 (21–59)	0.3^a^
AST (U/L)	31.5 (14–850)	25 (20–41)	**0.02^a^**
Cholesterol	108.69 ± 36.44	125.56 ± 32.21	0.2^b^
TG (mg/dL)	58.83 ± 19.4	45.67 ± 11.96	**0.06^b^**

a—Mann–Whitney, b—T-student. The *p*-values marked in bold are significant.

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
