# Peer review of "The Role of MECP2 and CCR5 Polymorphisms on the Development and Course of Systemic Lupus Erythematosus"

_biomolecules, 2020, doi:10.3390/biom10030494_

Round 1

Reviewer 1 Report

This paper measures MECP2 and CCR5 polymorphisms in whole blood samples from patients with lupus.  The paper is limited by the following factors:

  1. No mention is made as to whether the patients met the standard requirements for the classification of lupus

  1. Table 2 lists the P value of every gene analyzed as well as the statistical significance, but only 11 meet the standard criteria of p <05. 

  1. There is no analysis as to whether the polymorphism affects the level of gene expression or affects the function of the associated protein.

Author Response

Reviewers' comments and action taken:

Response to Reviewer 1

This paper measures MECP2 and CCR5 polymorphisms in whole blood samples from patients with lupus. The paper is limited by the following factors:

Comment 1. No mention is made as to whether the patients met the standard requirements for the classification of lupus

Thanks’ the Reviewer for comments. In Materials and Methods 2.1. Study group section we have information “All patients with SLE included in the study met the classification criteria of American College of Rheumatology (ACR) for this disease”, however, we also add this information: “SLE patients included in the study have the presence of at least four criteria”. Please refer to the revised version of the manuscript.  

Comment 2. Table 2 lists the P value of every gene analyzed as well as the statistical significance, but only 11 meet the standard criteria of p<05.

Thank you very much for the Reviewer's kind comments. All p-value which were not significant we corrected as NS. In table 2 we leave only significant p-value and p-value with a tendency. Please refer to the revised version of the manuscript. 

Comment 3. There is no analysis as to whether the polymorphism affects the level of gene expression or affects the function of the associated protein.

We deeply appreciate the Reviewer’s professional remarks. Unfortunately, at this moment, we don’t have information on examined gene expression in our study group. Moreover, in this moment we are unable to do these analysis because we do not have a reagents. But we agree with the reviewer that these analysis may be important and we would like continue our research in this subject. We also did the bioinformatic analysis to check  if examined SNPs affect the function of the associated protein. Computational functional prediction analysis was performed using MutationTaster and PROVEAN programs. The analysis showed that not MECP2 SNPs, but deletion of 32 bp in rs333 CCR5 might result in altered function of the associated protein. The amino acid sequence is changed, what consequences in frameshift. It leads to changes in splice site and also truncate the protein (might cause nonsense-mediated mRNA decay (NMD)).

We added information about software used to perform biostatistical analysis in Statistical analysis subsection as well as information about outcomes of biostatistical analysis in MECP2 and CCR5 polymorphisms and SLE susceptibility subsection. We added the two references in the Reference section. Please refer to the revised version of the manuscript. 

Neither the entire work nor any part of its content has been published or accepted elsewhere. The authors declare no conflict of interest. All authors read and accepted revised version of the paper.

                       Your sincerely,

                       Agnieszka Paradowska-Gorycka, PhD

                       Department of Molecular Biology

                       National Institute of Geriatrics, Rheumatology and Rehabilitation

                       Poland, Warsaw

Reviewer 2 Report

Here the authors reported targeted genotyping study of SLE patients. Improper statistical analysis and citations raise question on the quality of this paper. Although some part of the findings could be important, this paper is too preliminary to be published in its present form. 

#major points

#1. What is the exact number of SLE patients? In the abstract, there is 604 SLE patients, but in the data, around 100-200 patients.

#2. Improper References. The references were cited by number in the article, such as [1]. However, in the references section, they are not numbered. In addition, in the method section, an article is cited with its PMID number. It is important to appropriately cite references in scientific research papers in order to acknowledge previous works in this field.

#3. Statistical analysis. P values “around” 0.05 was considered significant in this analysis. This ambiguous cut-off value is not scientifically sound. Considering multiple testing of this article, even p-values of 0.05 is not significant enough. When performing genome wide association studies, p-value of 5.0 x 10-8 is classically considered significant based on the idea of Bonferroni method. In this standpoint, I recommend to reconsider statistical significances of this article. Use of FDR might help.

#4. SNPs on focusd MECP2 and CCR5 genes were selected by PubMed search. Do these selected SNPs include all common SNPs in European or Polish populations?

#5. I believe that the genome wide association study (GWAS) is the most robust statistical method to identify trait associated variants. Bentham J et al (PMID: 26502338) reported MECP2 rs1734787, not CCR5 SNPs in the European lupus GWAS. Discussion on these GWAS papers will help the readers.

Author Response

Reviewers' comments and action taken:

Response to Reviewer 2

Here the authors reported targeted genotyping study of SLE patients. Improper statistical analysis and citations raise question on the quality of this paper. Although some part of the findings could be important, this paper is too preliminary to be published in its present form.

Comment 1. What is the exact number of SLE patients? In the abstract, there is 604 SLE patients, but in the data, around 100-200 patients.

Thank you for the Reviewer’s comments. We corrected this information in Abstract and in the Material and Methods section. In this study we had a 137 SLE patients and 604 healthy subjects. Please refer to the revised version of the manuscript. 

Comment 2. Improper References. The references were cited by number in the article, such as [1]. However, in the references section, they are not numbered. In addition, in the method section, an article is cited with its PMID number. It is important to appropriately cite references in scientific research papers in order to acknowledge previous works in this field.

Thank you for the Reviewer’s comments. We corrected these information’s. We added the number of all used references in the Reference section. And we also deleted the PMID article from Method section. Please refer to the revised version of the manuscript. 

Comment 3. Statistical analysis. P values “around” 0.05 was considered significant in this analysis. This ambiguous cut-off value is not scientifically sound. Considering multiple testing of this article, even p-values of 0.05 is not significant enough. When performing genome wide association studies, p-value of 5.0 x 10-8 is classically considered significant based on the idea of Bonferroni method. In this standpoint, I recommend to reconsider statistical significances of this article. Use of FDR might help.

As Reviewer suggested we used Bonferroni correction. This information: ”Bonferroni correction was used to adjusting p-values for multiple testing; this correction was used to determine the association between examined gene polymorphisms and clinical phenotype of SLE. Bonferroni-corrected α-level of p < 0.003 was considered statistically significant” was added to the Statistical Analysis section. In the results section we also added some information about this correction. Please refer to the revised version of the manuscript.

Comment 4. SNPs on focusd MECP2 and CCR5 genes were selected by PubMed search. Do these selected SNPs include all common SNPs in European or Polish populations?

We deeply appreciate the Reviewer’s professional remarks. In the present study, for the first time, we analyzed MECP2 gene polymorphisms in the Polish population. CCR5 was examined in the Polish population, however, all studies examined only CCR5 rs333 genetic variants. For our analysis we decided chosen 4 MECP2 gene SNPs with the strongest association with SLE (due to financial constraints). There is more common MECP2 SNPs in European population than those selected by us. Webb et al. (2009) in their paper entitled“Variants within MECP2, a key transcriptional regulator, are associated with increased susceptibility to lupus and differential gene expression in lupus patients” and Sawalha et al. (2008) in“Common Variants within MECP2 Confer Risk of Systemic Lupus Erythematosus” examined MECP2 SNPs (all with MAF above 5% in European population). Webb et al. studied four MECP2 SNPs-  rs2075596, rs1734787, rs17435, rs2239464 (selected also by us) as well as: rs3027933, rs3027935, rs3027939, rs7050901, rs1624766, rs7884370, rs5987201, rs1734791, rs1734792, rs11156611. After analysis only rs2075596, rs3027933,  rs17435, rs1624766, rs1734787, rs1734791, rs1734792 and rs2239464 remain statistically significant (with p values as follows: 5.66×10−5; 1.50×10−5; 0.0027; 0.0038; 5.22×10−5; 1.92×10−5; 2.80×10−5; 0.001).

Sawalha et al. examined four MECP2 SNPs-  rs2075596, rs1734787, rs17435, rs2239464 (selected also by us) as well as: rs2266890, rs3027933, rs3027935, rs3027939, rs1624766,  rs7884370, rs5987201, rs1734791, rs1734792, rs11156611, rs5945175. After analysis only   rs2266890, rs2075596, rs3027933, rs17435, rs1624766, rs1734787, rs1734791, rs1734792 and rs2239464 remain statistically significant remain statistically significant (with p values as follows: 0.013; 0.0034; 0.0017; 0.0008; 0.0013; 0.0008; 0.0012; 0.0014; 0.0044).

In discussion we added information “First is limited number of SNPs, which were examined in this study.”. Please refer to the revised version of the manuscript.

Comment 5. I believe that the genome wide association study (GWAS) is the most robust statistical method to identify trait associated variants. Bentham J et al (PMID: 26502338) reported MECP2 rs1734787, not CCR5 SNPs in the European lupus GWAS. Discussion on these GWAS papers will help the readers.

Thank you very much for the Reviewer's comment. As Reviewer suggested we added abovementioned article to references and discussion.

In discussion we added information “Bentham et al. revealed in his GWAS analysis that MECP2 rs1734787 is an appropriate SLE risk factor for people of European ancestry.”. Please refer to the revised version of the manuscript.

Neither the entire work nor any part of its content has been published or accepted elsewhere. The authors declare no conflict of interest. All authors read and accepted revised version of the paper.

                        Your sincerely,

                       Agnieszka Paradowska-Gorycka, PhD

                       Department of Molecular Biology

                       National Institute of Geriatrics, Rheumatology and Rehabilitation

                       Poland, Warsaw

Round 2

Reviewer 1 Report

This paper is now suitable for publication.

Reviewer 2 Report

All of my comments have been sufficiently addressed in this revised manuscript.